# Prospective Risk of Type 2 Diabetes in Normal Weight Women with Polycystic Ovary Syndrome

**DOI:** 10.3390/biomedicines10061455

**Published:** 2022-06-20

**Authors:** Dorte Glintborg, Naja Due Kolster, Pernille Ravn, Marianne Skovsager Andersen

**Affiliations:** 1Department of Endocrinology, Odense University Hospital, DK-5000 Odense C, Denmark; najakolster@gmail.com (N.D.K.); msa@rsyd.dk (M.S.A.); 2Department of Gynecology and Obstetrics, Odense University Hospital, DK-5000 Odense C, Denmark; pernille.ravn@rsyd.dk

**Keywords:** polycystic ovary syndrome, type 2 diabetes, prospective, review

## Abstract

Background: Polycystic ovary syndrome (PCOS) is associated with obesity and increased risk for type 2 diabetes (T2D). However, the prospective risk of T2D in normal weight women with PCOS is debated, together with the relevant prospective screening programs for T2D in normal weight women with PCOS. Aim: To review and discuss prospective risk of T2D in normal weight women with PCOS, and to give recommendations regarding prospective screening for T2D in normal weight women with PCOS. Methods: Systematic review. Results: A systematic literature search resulted in 15 published prospective studies (10 controlled studies and 5 uncontrolled studies) regarding risk of T2D in study cohorts of PCOS, where data from normal weight women with PCOS were presented separately. In controlled studies, higher risk of T2D in normal weight women with PCOS compared to controls was reported in 4/10 studies, which included one study where T2D diagnosis was based on glucose measurement, two register-based studies, and one study where diagnosis of T2D was self-reported. Six of the 10 controlled studies reported no increased risk of T2D in normal weight women with PCOS. Four of these studies based the diagnosis of T2D on biochemical measurements, which supported the risk of surveillance bias in PCOS. In uncontrolled studies, 2/5 reported a higher risk of T2D in lean women with PCOS compared to the general population. We discuss the evidence for insulin resistance and β-cell dysfunction in normal weight women with PCOS, and aggravation in the hyperandrogenic phenotype, ageing women, and women with Asian ethnicity. Impaired glucose tolerance could be an important metabolic and vascular risk marker in PCOS. Conclusions: The risk of T2D may be increased in some normal weight women with PCOS. Individual risk markers such as hyperandrogenism, age >40 years, Asian ethnicity, and weight gain should determine prospective screening programs in normal weight women with PCOS.

## 1. Introduction

Polycystic ovary syndrome (PCOS) is a common endocrine disorder of reproductive-aged women, with a prevalence of 15–20% [1]. Polycystic ovary syndrome (PCOS) is most often defined according to the Rotterdam criteria, which include irregular ovulation, biochemical/clinical hyperandrogenism, and/or polycystic ovaries, when other causes are excluded [2,3]. The etiology of PCOS is considered multifactorial [4], but insulin resistance is a central part of the pathogenesis [5]. Insulin resistance is closely associated with obesity [5,6], and obesity is associated with increased risk of type 2 diabetes (T2D) in PCOS [7,8,9]. T2D was diagnosed in 1.5–10% women with PCOS, which corresponded to an odds ratio of 4.4 compared with controls [4].

Normal weight women constitute about 25% of women with PCOS [10]. Normal weight is usually defined by BMI < 25 kg/m^2^ or BMI < 23 kg/m^2^ in women of Asian origin [11]. In previous studies, abdominal fat mass was the most important predictor for metabolic risk in PCOS [12,13]. A recent meta-analysis found that the comparable risk for T2D in normal weight women with PCOS vs. controls was 3.34 (0.03; 400.5, *p* = 0.62), but the paper included cross-sectional as well as prospective studies [14]. A recent systematic review on prospective studies in PCOS [15] reported a higher incidence of T2D, but the authors did not present results from normal weight women separately [15]. The risk of T2D is increased in women with previous gestational diabetes mellitus; however, two recent Nordic studies reported no increase in gestational diabetes mellitus in normal weight women with PCOS [16,17]. A population-based Finnish study reported a synergistic effect of overweight/obesity and PCOS for prospective risk of T2D; whereas, the risk of T2D was not increased in normal weight women with PCOS [8]. If obesity is the primary risk factor for development of T2D in PCOS [8,9] needs clarification. Recommendations for prospective screening for T2D in normal weight women with PCOS may be different from obese women with PCOS.

PCOS phenotype could modify risk for T2D. Women with PCOS and hyperandrogenism could have a more adverse metabolic phenotype than normoandrogenic women [18], but the data were not uniform [19,20]. The pathophysiology of T2D involves β-cell dysfunction and insulin resistance. Increasing age is associated with loss of β-cell function [21], and age > 40 years is considered a separate risk factor for T2D [19,22].

Baseline screening for T2D in all women with PCOS is generally recommended, whereas guidelines differ regarding recommendations for follow-up after PCOS diagnosis [19,23]. Prospective screening in all women with PCOS is recommended by The Endocrine Society and the Androgen Excess and Polycystic Ovary Syndrome Society (AE-PCOS) and by the evidence-based guidelines on PCOS diagnosis and management developed in Australia [19]. Screening depending on baseline risk factors, such as high BMI (>30 kg/m^2^), age > 40 years, and family history of T2D, is recommended by the European Society of Human Reproduction and Embryology and the American Society of Reproductive Medicine [19].

The aim of this study was to investigate the evidence regarding development of T2D in normal weight women with PCOS and the possible moderating effect of PCOS phenotype including hyperandrogenism, age, and ethnicity. Due to the limited evidence, we included a discussion regarding mediating factors for insulin resistance and β-cell dysfunction in PCOS.

## 2. Materials and Methods

The research question regarding prospective risk for T2D in normal weight women with PCOS was investigated using a systematic literature search, as described below. The importance of PCOS phenotype for risk of T2D was investigated in the retrieved papers and further addressed using data from cross-sectional studies in PCOS. Recent papers regarding mechanisms for insulin resistance and β-cell dysfunction in PCOS were included in the discussion.

We searched for studies published in PubMed and Embase until April 2022. The terms used in the search were polycystic ovary syndrome, polycystic ovarian syndrome, PCOS, diabetes, T2D, T2DM, NIDDM, and HbA1c. Furthermore, a lateral search was conducted based on reference lists of included articles. Recent meta-analyses regarding PCOS and T2D were hand-searched for potentially eligible articles. Controlled studies were defined as clinical studies with a comparison (control) group. The following PECOs (population, exposure, comparator, and outcome) were included: population: pre- and postmenopausal women; exposure: PCOS; comparator: healthy control or background population; outcome: T2D. Criteria for study selection are further specified below.

### 2.1. Inclusion Criteria

Publications had to include prospective studies on development of T2D in women with PCOS. The definitions of PCOS were by Rotterdam criteria or by National Institute of Health (NIH) [2,3], according to diagnosis code (ICD-10) or could be self-reported. Study cohorts had to include normal weight women with PCOS or cohorts of mixed BMI, where normal weight women with PCOS were investigated separately as a sub-cohort. Studies without a control cohort were allowed, but the results of these studies were presented and discussed separately.

### 2.2. Study Outcome

The primary study outcome was diagnosis of T2D based on WHO criteria using HbA1c or oral glucose tolerance test (OGTT) [24], or alternatively self-reported diagnosis of T2D or diagnosis code on T2D in register based studies.

### 2.3. Exclusion Criteria

Excluded publications were cross-sectional studies, intervention studies, studies in pregnant women, reviews, and editorials. Studies in cohorts of mixed BMI were excluded if they did not report results in normal weight women separately.

### 2.4. BMI and Body Composition

We applied BMI categories according to the WHO, where underweight is defined as BMI ≤ 18.4 kg/m^2^, normal weight 18.5–24.9 kg/m^2^, overweight 25–29.9 kg/m^2^, and obesity ≥30 kg/m^2^ [11]. In Asian populations, normal weight was defined as BMI < 23 kg/m^2^ [11]. Most studies used the term lean for the combined groups of underweight and normal weight women.

We retrieved information regarding first author, publication year, country of origin, study design, study population (age, ethnicity), PCOS definition, follow-up, and study results (Table 1). The applied definition of normal weight is presented in Table 1.

## 3. Results

### 3.1. Study Selection and Descriptive Data

The process of study selection is shown in Figure 1. A total of 3292 studies were identified and screened. Based on a priori selection criteria screening of title and abstract, 3134 studies were excluded, leaving 158 for assessment of the full text and 15 studies for final inclusion. Selection of studies was crosschecked by evaluation of reference lists of three recent reviews regarding metabolic outcomes in PCOS [7,14,15], which did not add additional references. Two publications included data from the Australian Longitudinal Study on Women’s Health [25,27], which implied overlapping study cohorts. The paper by Glintborg [29] provided additional data to the paper from the same study group [9], and the results were presented as one publication [29]. The paper by Persson [18] discussed results from normal weight women, but these data were not shown in the paper and were retrieved after personal communication with the senior author (Poromaa IS). Characteristics of included studies are presented in Table 1.

The studies were published 2010–2021 and performed in European [8,9,18,29,34,35,37], Northern American [31,32], Asian [26,28,30,33], and Oceanic [25,27] study cohorts. No study was conducted in Southern American or African study cohorts. All studies included study cohorts of mixed BMI, and no study included normal weight participants only. Most studies used the term lean for the combined groups of underweight and normal weight women. The number of lean or normal weight women with PCOS ranged from *n* = 14 to 12,362; but in several studies, the size of the lean study cohort was not presented separately (Table 1). The study by Ryu [26] was conducted in a Korean study population, but the study defined non-obese as BMI < 25 kg/m^2^. The study by Boudreaux [32] did not present data from normal weight women with PCOS but found the discriminatory cut-off value BMI >35 kg/m^2^, below which the risk of T2D was not increased. Follow-up duration ranged from 1 to 32 years (Table 1). Ten studies included a control group (upper panel of Table 1), whereas the remaining five studies compared risk of T2D in women with PCOS to findings in the general population or within the PCOS group (lower panel of Table 1).

The definition of PCOS was based on clinical evaluation [28,29,30,32,33,34,35,37], ICD10 diagnosis code [18,26], or self-reported data from questionnaires or interviews [8,25,26,31]. Wang [31] applied questionnaires regarding menstrual status in combination with measurement of serum testosterone levels.

Diagnosis of T2D was based on OGTT [8,28,33,34,35,37] or fasting blood glucose [31,32,35], ICD-10 diagnosis code [18,27,33], or self-reported [25,27]. Medical treatment for T2D (excluding metformin) was added to ICD10 diagnosis code by two studies [18,29]. T2D was defined by American Diabetes Association (ADA) [28,30,31,32,33,34,35,36] or WHO criteria [8,37]. Prediabetes was included in the primary study outcomes by one study [33].

### 3.2. PCOS and Prospective Risk of T2D

#### 3.2.1. Controlled Studies

Four controlled studies reported a higher risk of T2D in lean/normal weight women with PCOS compared to controls [18,26,31,33], whereas comparable risk for T2D in PCOS and controls was reported in the remaining six studies [8,25,27,32,34,35]. Studies reporting higher risk for T2D in PCOS included study populations from Sweden [18], Korea [26], Australia [27], and USA [31]. Biochemical assessment was applied for diagnosis of T2D in one study only [31], whereas the three other studies used a diagnosis code [18,26], medicine prescription [18], and self-reported diagnosis [27] of T2D.

Persson [18] reported adjusted hazard ratios (HR) (95% CI) for T2D of 2.01 (1.29–3.12) and 4.27 (2.60–7.00) in lean women with PCOS, and without and with hyperandrogenism, respectively, vs. controls. The study was register-based, and information on BMI was missing in 54% women with PCOS. Use of antiandrogen oral contraceptives or antiandrogen drugs was used to define the hyperandrogen phenotype of PCOS.

Ryu [26] reported adjusted HR (95% CI) for T2D 2.3 (1.7–3.2) during 4.5 years follow-up in non-obese women with PCOS vs. non-obese controls. The authors applied a cut-off for non-obese at BMI 25 kg/m^2^, which is higher than the recommended cut-off <23 kg/m^2^ in Asian study populations [26]. Register-based data was applied to obtain the diagnoses of PCOS and T2D [26].

The study by Kakoli [27] reported an incidence risk rate (IRR) (95% CI) of 4.68 (2.66–7.91) in lean women with PCOS vs. lean controls during mean follow-up of 15 years. PCOS and T2D were self-reported using questionnaires [27].

Wang [31] reported OR for T2D (95% CI) 3.1(1.2–8.0) in *n* = 31 lean women with PCOS vs. controls during 18 years follow-up. The diagnosis of PCOS was based on questionnaires and testosterone measurement, and the diagnosis of T2D was based on fasting plasma glucose [31].

Six controlled studies reported no increased risk for T2D in lean women with PCOS vs. controls [8,25,27,32,34,35]. Four of these studies based the diagnosis of PCOS on clinical examination [27,32,34,35], and two studies applied questionnaire data [8,25] or diagnosis codes [8]. The diagnosis of T2D was based on biochemical measurements in five studies: OGTT [8,28] and fasting plasma glucose [31,32,35], whereas two studies applied self-reported T2D [25] and ICD-10 diagnosis codes and medicine prescriptions [29].

#### 3.2.2. Uncontrolled Studies

Two uncontrolled studies reported higher risk for T2D in lean women with PCOS compared to the general population [33,37], and three studies found no increased risk for T2D in PCOS [28,29,37]. All uncontrolled studies applied clinical examination for diagnosis of PCOS, and the diagnosis of T2D was based on OGTT [33,34,35,36,37].

The two studies showing higher risk for T2D in lean women with PCOS were conducted in South Korean [33] and Danish [37] study cohorts. Choi [33] found incidence rate of 5.5/1000 person years in 212 women with PCOS and waist < 80 cm, which was significantly higher than the reported incidence rate of 0.9/1000 person years for T2D in 15,050 women aged 20–29 years in the Korean National Health Insurance Database.

Andries [37] reported T2D in 3/36 women with hyperandrogen phenotype during an average follow-up of 4 years, which was higher than that expected in the healthy controls (expected results not presented).

Studies reporting no increased risk for T2D in lean women with PCOS [28,29,37] were conducted in Polish [34], Bosnian [35], and Italian [36] study cohorts. The studies reported no incident cases of T2D during follow up [34,35] or very low incidence rate [36], which was not higher than in the background population. The study cohort of Jacewicz-Święcka [34] included 14 lean women with PCOS.

#### 3.2.3. PCOS Phenotype and Risk of T2D

Predisposing factors for development of T2D were investigated in several studies [8,18,25,27,28,29], but only one study presented analyses restricted to lean women separately [18], and results were retrieved by personal communication. Persson [18] reported that the hyperandrogenic phenotype was an independent predictor of T2D in the sub-cohort of lean women with PCOS [18]. The HR was 4.27 (2.60; 7.00) in lean hyperandrogenic women with PCOS vs. controls and 2.01 (1.29; 3.12) in lean normoandrogenic women with PCOS vs. controls [18]. The study was register-based and did not include biochemical measurements. Hyperandrogenism was defined according to prescription of antiandrogen oral contraceptives and/or prescription of antiandrogen drugs [18]. In addition, older age, non-Caucasian origin, and short education were independent predictors for T2D in lean women with PCOS [18]. The highest HR for T2D was reported in women originating from India, Pakistan, or Bangladesh: HR 8.45 (4.11; 17.39) compared to women originating from Nordic countries [18]. Study results by Persson [18] supported similar predictors of T2D risk in lean and obese women with PCOS. Higher BMI, age, and ethnicity were also significant predictors of T2D in other studies [8,25,27,28,29], whereas results regarding hyperandrogenic phenotype were not uniform [28,29]. Ollila [8] found that the weight increase between 14 and 31 years was significantly higher in women with PCOS, who developed T2D, compared to women with PCOS and normal glucose tolerance, median (25%; 75% quartiles): 27.3 (20.4; 34.8) kg vs. 13.8 (8.6; 20.2) kg, *p* < 0.001). Ng [28] found a 1.7-times higher incidence rate of T2D in women with PCOS and hyperandrogenism compared to women without hyperandrogenism. However, this association became non-significant after adjustment for BMI [28]. Glintborg [9,29] found that baseline fasting blood glucose, 2-h blood glucose during OGTT, and triglycerides were the best predictors of development of T2D, whereas serum testosterone levels and PCOS phenotype did not predict T2D. Accordingly, Choi [33] observed that the incidence rate of T2D was significantly higher (*p* = 0.002) in women with fasting plasma glucose ≥ 5.6 mmol/L at baseline than in women with fasting plasma glucose < 5.6 mmol/L.

Irregular menstrual cycle was not more prevalent in women with PCOS and prediabetes compared to women with PCOS and normal glycemic status in the studies by Velija-Asimi [35] and Choi [33]. Kiconco reported that menstrual regularity was not an independent predictor for diabetes at multivariable analysis level, where BMI was included in the regression model [25]. Wang [31] found that 26/41 women had spontaneous resolution of PCOS symptomatology during follow-up, which included regular menstrual cycles. Women with resolution of PCOS symptoms did not have a higher risk of T2D compared to controls [31].

## 4. Discussion

In the present paper we discuss the evidence regarding prospective risk for T2D and possible predictors of T2D in normal weight women with PCOS. The number of publications was limited, and no study was conducted in a cohort only consisting of normal weight or lean women with PCOS.

### 4.1. Diagnosis of PCOS and Surveillance Bias

Interestingly, the majority of evidence supporting higher risk of T2D in normal weight women in PCOS came from register-based studies where the diagnosis of T2D was based on ICD10 diagnosis codes [18,26] and medicine prescription [18], or studies where the diagnosis of T2D was self-reported [27]. These studies included a relatively high number of women with PCOS (>500), and several studies had a follow-up duration up to 20 years [18,26,33]. According to many international guidelines, women with PCOS are recommended annual metabolic risk screening [19]. Symptoms of T2D are often mild or absent, and diagnosis may be postponed for several years or remain unrecognized in women not attending regular screening programs. In the general population, the proportion of undiagnosed T2D is 25–50% [38,39]. Systematic screening for T2D in PCOS may introduce surveillance bias in register-based studies comparing development of T2D in women with PCOS vs. controls. Choi [33] found higher risk for T2D in a South Korean study population of lean women with PCOS (mean BMI 22.7 kg/m^2^) compared to the general population; however, the risk for T2D in the general population could be underestimated, which would affect study conclusions. The register based study by our own group [29] found that risk of T2D was not increased in 421 lean women with PCOS. T2D was diagnosed by register data and medicine prescriptions of antidiabetic treatment (excluding metformin), which could induce surveillance bias [29]. However, Danish national guidelines do not recommend prospective metabolic screening in lean women with PCOS, which is likely to have resulted in a similar follow-up of women with PCOS and controls [29]. The diagnosis of T2D was based on biochemical assessment in several studies [8,28,30,31,32,33,34,35,36,37], which would imply a similar evaluation at follow-up in patients and controls and, therefore, less risk of surveillance bias compared to register data and self-reported data. Interestingly, the majority of these studies did not report increased risk for T2D in lean women with PCOS [8,28,29,32,34,35,37], but many studies were limited by a small sample size, and different definitions were applied for T2D diagnosis. The largest clinical study cohort with the longest follow-up was the Finnish birth cohort, including 62 lean women with self-reported PCOS [8]. Follow-up duration was 32 years, and no lean women with PCOS had T2D at the age of 46 years based on OGTT [8]. The authors concluded that only overweight/obese women with PCOS should be screened for T2D [8]. Similarly, three controlled studies [32,34,35] and three uncontrolled studies [28,29,37] with biochemical diagnosis of T2D reported no increased risk of T2D in lean women with PCOS.

### 4.2. Hyperandrogenism and Metabolic Risk in PCOS

Wang [31] reported a higher risk of T2D in 21 lean women with PCOS compared to controls. Wang applied the NIH criteria for PCOS, which implied clinical and/or biochemical hyperandrogenism in all the included women [31]. Interestingly, a higher metabolic risk in hyperandrogenic women with PCOS was supported by Andries [37], where 3/36 lean women with PCOS and/or hirsutism developed T2D during follow-up. These findings support that the hyperandrogenic phenotype of PCOS could be associated with a higher risk for T2D [31,37]. Persson [18] found that hyperandrogenism was an independent predictor of T2D risk in the sub-cohort of lean women with PCOS. In accordance, patients fulfilling the NIH criteria for PCOS had a more adverse metabolic risk profile than patients with milder phenotypes [40], and hyperandrogenism could aggravate insulin resistance [18,41]. These findings contrasted other studies, where PCOS phenotype and hyperandrogenism did not predict T2D risk [28,29] (Figure 2). 

### 4.3. Menstrual Cycles and Metabolic Risk in PCOS

Irregular menstrual cycles is associated with insulin resistance and central obesity [42,43,44]. Wang [31] found that women with PCOS and a normalized menstrual cycle during follow-up did not have higher risk of T2D compared to controls [31]. Kiconco reported that irregular menstrual cycle did not predict risk of T2D [25], but correction for BMI in regression analyses could possibly explain these findings. Irregular menses could be a valuable tool to indicate increased risk of T2D in women with PCOS.

### 4.4. Oral Contraceptives and Metabolic Risk in PCOS

In our register based study, prescription of oral contraceptives was associated with higher risk of T2D (HR = 1.4; 95% CI, 1.3 to 1.6) [9,29]. Treatment with oral contraceptives regulates menstrual cycle and increases SHBG levels, leading to decreased levels of free testosterone and decreased hirsutism scores [45]. More than 50% of Danish women with PCOS had prescriptions of OCP compared to 25% controls [46]. Second generation oral contraceptives are considered first choice due to lower thromboembolic risk, but first generation OCPs containing drospirenone may be superior regarding antiandrogen effects [45,47]. The use of antiandrogenic oral contraceptives or antiandrogenic drugs was used to define hyperandrogenism in the study by Persson [18], and this indirect definition of hyperandrogenism may have influenced the study outcomes regarding higher risk of T2D in hyperandrogenic women with PCOS compared to the non-hyperandrogenic phenotype. The study by Persson relied on the assumption that a 4th generation oral contraception is used specifically in women with PCOS and hyperandrogenism [18]. Their study design could not account for use of 4th generation oral contraceptives for treatment of menstrual irregularity unrelated to hyperandrogenism [18]. Oral contraceptives [48], and especially oral contraceptives containing cyproterone acetate as progestin and antiandrogen, could have adverse effects on insulin resistance [49,50]. In obese women with PCOS using this oral contraceptive, Morin-Papunen [50] reported a higher area under the curve for glucose during OGTT. In lean women with PCOS, Christakou [51] compared oral contraceptives with cyproterone acetate, oral contraceptives with drospirenone, or metformin. During six months treatment, BMI and HOMA-IR were higher in both groups treated with oral contraceptives [51]. Weight gain during oral contraceptive treatment, and also in lean women with PCOS, [48,51] could increase the risk of development of T2D and more data are needed regarding weight change and risk of T2D during oral contraceptive treatment in normal weight women with PCOS [45].

### 4.5. Metformin and Myoinositol in PCOS

Metformin increases insulin sensitivity and improves ovulatory function in PCOS, whereas androgen levels and hirsutism scores are only mildly improved [52]. Metformin is widely used in PCOS, and we previously reported that 12% Danish women with PCOS had a prescription for metformin compared to 0.4% controls [46]. In our previous randomized controlled trial [48], 12 months metformin treatment induced a median weight loss of 3 kg in a normal weight study population (median BMI 25.1 kg/m^2^) with PCOS. In accordance, Christakou [51] reported a BMI decrease from 23.0 to 22.4 kg/m^2^ and lower CRP levels during six months metformin treatment in lean women with PCOS. This supported the metabolic benefit of metformin treatment in normal weight women with PCOS [48]. Furthermore, metformin could have anti-inflammatory effects [13] and metformin also improved vascular markers in lean women with PCOS [48,51]. Gastrointestinal side effects may, however, limit long-term compliance to metformin [48]. More recently, myoinositol was introduced as an insulin sensitizer in PCOS [53]. Myoinotisol treatment compared to placebo improved the insulin sensitivity in women with PCOS, without significant effects on BMI [53,54]. Myoinositol has predominantly been used as part of fertility treatments in PCOS, and long-term data are missing. In young lean women with PCOS, three months of myoinositol prevented weight gain and increased insulin sensitivity [55], and additional studies are needed to determine the possible benefits of myoinositol, regarding risk of T2D in normal weight women with PCOS.

### 4.6. Autoimmunity and Hypovitaminosis D

Autoimmunity is increased in PCOS, due to higher inflammatory status, unbalanced estrogen/progesterone secretion, hypovitaminosis D, or still unknown mechanisms [4,56]. Women with PCOS had higher secretion of anti-thyroid, anti-nuclear (ANA), anti-ovarian, and anti-islet cell antibodies [56]. The risk of overt thyroid disease is increased 3.6 times in PCOS vs. controls [46], and subclinical hypothyroidism is present in 10–25% women with PCOS [57]. In PCOS, subclinical hypothyroidism was linked to a more disadvantageous lipid profile and higher HOMA-IR [57,58], but glucose levels during OGTT were comparable [57]. Studies on long-term morbidity in women with PCOS and subclinical hypothyroidism are needed [57]. Screening for overt thyroid disease is part of the recommended evaluation at baseline in all women with PCOS, but the impact of subclinical hypothyroidism for T2D risk remains to be determined [58]. Hypovitaminosis D may be present in more than 75% women with PCOS [59]. Vitamin D levels were inversely associated with BMI [60], and risk of hypovitaminosis D in normal weight women may be determined by fat mass [61]. Treatment with vitamin D in PCOS was associated with decreased HOMA-IR and glucose levels [62]. Therefore, screening for vitamin D levels is relevant upon diagnosis of PCOS, and vitamin D substitution should be considered in women with low vitamin D levels.

### 4.7. Depression

The risk of depression was increased up to five times in PCOS [46,63], and 16.9% of Danish women with PCOS had prescriptions for antidepressants compared to 8.8% of controls [46]. Depression is associated with higher risk of T2D [64], and use of antidepressants could further increase the risk of T2D [65]. We recently reported that treatment with escitalopram increased waist circumference in women with PCOS and no clinical depression, whereas measures of insulin resistance were unchanged [66]. Women with PCOS and depression could be at risk for T2D, especially in the case of weight gain.

### 4.8. Ageing and β-Cell Function in PCOS

Older age predicted risk of T2D in lean women [18] and mixed study cohorts of PCOS [8,25,27,34]. T2D develops due to decreased β-cell function and increased insulin resistance [67]. β-cell function declines with advancing age [21], and gradual reduction of β-cell function can result in progression from normal glucose tolerance to T2D. Furthermore, increasing age is associated with higher BMI and waist circumference [68], and all factors deteriorate insulin sensitivity [12]. In accordance, Ollila [8] reported that weight gain was an important predictor of the development of T2D. In the study by Jacewicz-Święcka [34], all women who developed prediabetes had a waist circumference ≥ 80 cm at follow-up. Metabolic risk increased proportionally with increasing BMI [8], implying a synergistic effect of increasing BMI in PCOS and the importance of early intervention to maintain healthy weight.

### 4.9. Pre-Diabetes and Method for T2D Diagnosis

Prediabetes is an important independent risk factor for diseases associated with T2D, such as cardiovascular disease, neuropathy, and nephropathy [67]. The presence of prediabetes may, therefore, indicate the need for intensified prospective screening for T2D, as glucose levels can revert to a normal range after a lifestyle intervention [69]. In accordance, several studies supported that higher fasting glucose levels predicted the risk of T2D [9,27,36]. How to assess prediabetes and T2D in PCOS has been debated [70]. OGTT is applied in some clinics, to diagnose T2D in women with PCOS, whereas existing guidelines consider HbA1c the preferred tool to diagnose T2D in the general population [24]. HbA1c compared to OGTT has low sensitivity for the diagnosis of prediabetes and T2D in PCOS [71], but HbA1c is a better predictor of cardiovascular disease and overall mortality than blood glucose during OGTT [71,72]. HbA1c could be used as an indicator for both increased metabolic and vascular risk [72,73], and prospective measurement of HbA1c is more feasible than measurement of fasting glucose or OGTT in a clinical setting. In some countries, the use of OGTT is preferable to HbA1c and fasting glucose, as other methods are unavailable or unreliable [70]. In normal weight women with PCOS, we suggest that screening for prediabetes and diabetes, and the indication for OGTT compared to HbA1c or fasting glucose, should resemble those in other patients at risk of T2D [24].

### 4.10. Diet Intervention in PCOS

The close association between obesity and metabolic risk in PCOS underlines the importance of lifestyle intervention as a first line treatment in women with obesity [74]. The optimal dietary approach as part of lifestyle management in PCOS remains controversial, with limited high-quality evidence to support any specific dietary approach for PCOS, beyond general population-based guidelines [75]. Prospective cohort studies in general populations showed that a high carbohydrate intake (more than 70%) could be associated with higher risk of T2D [76]. In PCOS, low carbohydrate intake (less than 55%) could decrease insulin resistance [77]. It is, however, still undetermined if normal weight women with PCOS can benefit from a specific diet composition, such as meals with low carbohydrate and high protein content. Lean women with PCOS had a later timing of breakfast and lunch than controls and a greater intake of junk food [78], but the impact of meal timing and composition for metabolic risk in normal weight women with PCOS is undetermined.

### 4.11. Ethnicity and T2D

Non-Caucasian origin was an independent predictor for T2D in lean women with PCOS [18,25,26]. Persson [18] reported the highest risk for T2D in women originating from India, Pakistan, or Bangladesh: HR 8.45 (4.11; 17.39) compared to women originating from Nordic countries. Higher risk for T2D in women from the Asian continent was supported by other studies [8,25,27,28,29]. A meta-analysis [79] found that women with PCOS had increased prevalence of T2D (OR = 2.87, 95% CI: 1.44–5.72) and IGT (OR = 3.26, 95% CI: 2.17–4.90), which differed by ethnicity (for IGT: Asia: 5-fold, America: 4-fold, and Europe: 3-fold). Ethnicity-related difference in metabolic risk remained significant for Asia and Europe in BMI-matched sub groups [79]. Consistently, women from East Asia with PCOS have the highest prevalence of metabolic syndrome, despite lower BMI and a milder hyperandrogenic phenotype [80]. A North American study of sub-groups of Asian women showed a higher risk of T2D in Filipino (adjusted OR 1.7) and South Asian (adjusted OR 2.3) women compared to Chinese women, after adjustments for age, BMI, and current smoking status, indicating a particular risk of metabolic syndrome in South Asian women [81]. Studies of genetic components and susceptibility loci for PCOS and single nucleotide polymorphism have shown variations between different ethnical groups [82]. At present, the data thus suggest that the risk of T2D across ethnic groups should be considered and aligned in the screening guidelines for general populations to facilitate early detection of dysglycemia [70,80].

## 5. Conclusions

We suggest that baseline metabolic screening should be performed in normal weight women with PCOS, according to existing guidelines. The available evidence did not support prospective screening at short intervals in young normal weight women with PCOS. Measurement of HbA1c or fasting glucose at, for example, five-year intervals in normal weight women with PCOS is suggested.

## Figures and Tables

**Figure 1 biomedicines-10-01455-f001:**
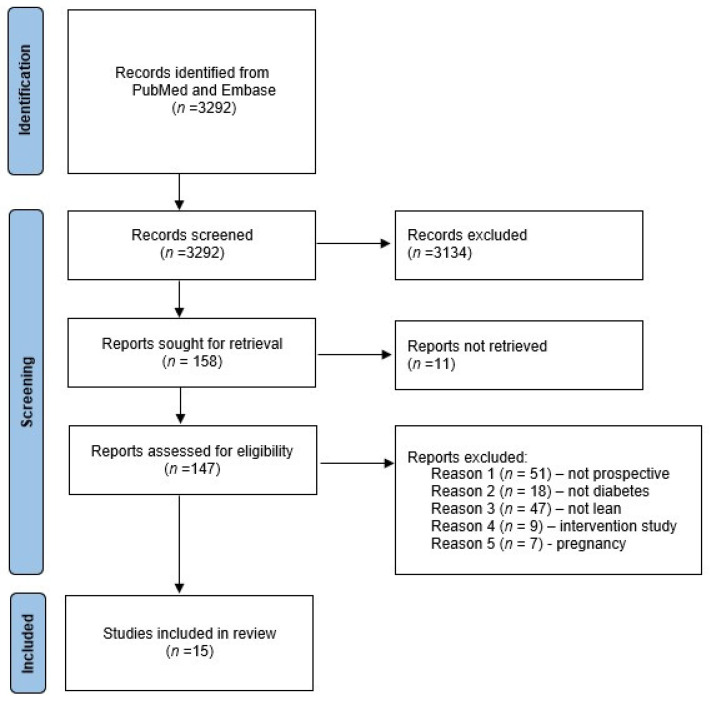
PRISMA flow chart of included studies regarding prospective risk of T2D in normal weight women with PCOS.

**Figure 2 biomedicines-10-01455-f002:**
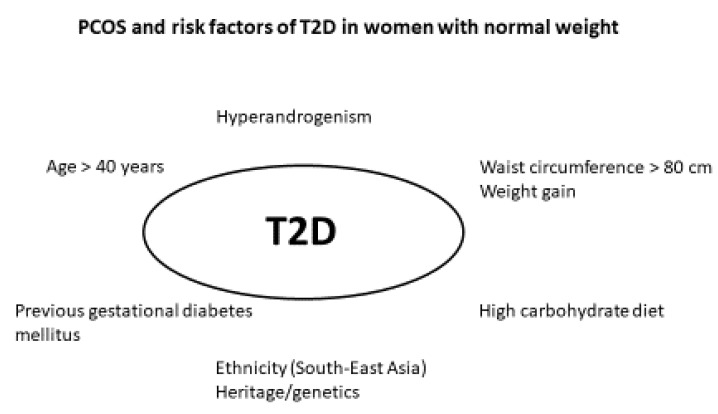
Risk factors for T2D in PCOS.

**Table 1 biomedicines-10-01455-t001:** Characteristics of included publications.

Study Descriptives		Total Population				Normal Weight Sub-Cohort			
Author, Year, Country	Design Study, Setting	Population (Number, Age, BMI (kg/m^2^))		PCOS Definition	Follow-Up Duration	Definition, BMI (kg/m^2^)	T2D Definition	Results	Conclusion T2D Normal Weight PCOS vs. Controls
		PCOS	Control						
Persson et al., 2021 [18]	Register-based	*n* = 52,535 (12,362 lean)	*n* = 254,624 (83,120 lean)	Diagnosis code (ICD-10)	Maximum 19 years	Lean < 25	Diagnosis code. Medicine prescriptions	HR 2.01 (1.29–3.12) normoandrogenic PCOS vs. controls, HR 4.27 (2.60–7.00) lean hyperandrogenic PCOS vs. controls	Increased risk of T2D in non-obese women with PCOS compared to non-obese controls. Hyperandrogenism independent risk factor in lean women with PCOS
Kiconco et al. 2021 Australia [25]	Prospective, birth cohort database	*n* = 1356 (lean N/A) Age 47.6 ± 1.5	*n* = 11,740 (lean N/A) Age 47.6 ± 1.5	Irregular menses (questionnaire)	20 years	Healthy weight 18.5–24.9	Self-reported T2D diagnosis	HR 0.95 (0.52–1.73) in healthy weight women with irregular menses vs. controls	Women with healthy weight did not have increased risk for T2D
Ryu et al., 2021 Korea [26]	Prospective, Population-based Register study	*n* = 1136 (818 lean) Age 15–44 (mean age not presented) BMI 21.79 ± 3.9	*n* = 5675 (4.546 lean) Age 15–44 (mean age not presented) BMI 21.06 ± 3.0	Diagnosis code (ICD-10)	4.5 years (2.4–6.2)	Non-obese < 25	Diagnosis code ICD10	Non-obese PCOS vs. non-obese controls: Adjusted HR (95% CI) 2.3 (1.7–3.2)	Increased risk of T2D in non-obese women with PCOS compared to non-obese controls.
Kakoly et al., 2019 Australia [27]	Prospective, Population-based	*n* = 707 (lean N/A)	*n* = 7671 (lean N/A)	Self-reported (questionnaire)	15 years	Lean < 25	Self-reported T2D diagnosis	Lean PCOS vs. lean controls IRR (95% CI): 4.68 (2.66–7.91)	Increased risk of T2D in lean women with PCOS compared to lean controls.
Ng et al., 2019 China [28]	Prospective, Hospital clinic- and community-based	*n* = 199 (lean N/A) Age 30.6 ± 6.5 BMI 25.9 ± 5.6	*n* = 225 (lean N/A) Age 42.6 ± 7.0 BMI 23.2 ± 3.8	Rotterdam Clinical evaluation	10.6 ± 1.3 years	Lean < 23	OGTT	Rate ratio PCOS vs. controls: 1.84 (0.65; 5.25)	No significant difference between lean women and controls.
Glintborg et al., 2018 Denmark [9,29]	Prospective, National register and hospital clinic	*n*= 18,477 Embedded local cohort: *n* = 1165 (n= 421 lean) Age 29 (22–35) BMI 27.0 (23.0–32.4)	*n* = 54.680 (lean N/A) Age 29 (23–35) Median BMI N/A	Rotterdam Clinical evaluation (local sub- cohort)	11.1 years (6.9–16.0)	Lean < 25	Diagnosis code Medicine prescriptions	Lean PCOS vs. controls:HR 1.22 (0.58; 2.55)	No increased risk of T2D in lean women with PCOS compared to age- and BMI-matched controls.
Ollila et al., 2017 Finland [8]	Prospective, Population-based cohort	*n* = 279 (*n* = 62 lean) Age 14 (Birth cohort 1966) BMI 28.6 ± 6.3 (end of study)	*n* = 1577 (559 lean) Age 14 (Birth cohort 1966) BMI 26.3 ± 5.3 (end of study)	Self-report. NIH or diagnosis of PCO/PCOS	32 years	Lean < 25	OGTT	Lean PCOS: OR for T2D: 1.10 (0.31–3.80) NS	No significant difference between lean women with PCOS and controls
Tehrani et al., 2015 Iran [30]	Prospective, Population-based	*n* = 85 (normal BMI N/A) Age 29.8 ± 9.2 BMI 27.2 ± 5.3	*n* = 552 (Normal BMI N/A) Age 29.3 ± 9.0 BMI 25.6 ± 5.0	NIH Clinical evaluation	9.4 years (8.7–10.4)	Normal BMI < 25	Self-reported diabetes and FPG	Normal BMI PCOS vs. controls: FPG: NS	No significant difference between normal BMI women with PCOS and controls
Wang et al., 2011 USA [31]	Prospective, Population-based	*n* = 53 (*n* = 31 lean) Age 26.8 ± 3.7 Mean WC 78.5 ± 13.9	*n* = 1074 (620 lean) Age 27.3 ± 3.6 Mean WC 77.0 ± 12.5	Self-reported + testosterone measurement	18 years	Lean < 25	FPG	Lean PCOS vs. lean controls: OR T2D: 3.1 (1.2–8.0)	Increased risk of T2D in lean women with PCOS compared to lean controls
Boudreaux et al., 2006 USA [32]	Prospective, Hospital clinic	*n* = 97 (lean N/A) Age 38 ± 5.9 BMI 31.6 ± 9.6	*n* = 95 (normal BMI N/A) Age 40 ± 5.2 BMI 26.22 ± 6.00	NIH Clinical evaluation	8 years	Lean < 25 (<35)	FPG	Women BMI < 35 kg/m^2^, PCOS vs. controls adjusted HR = 1.45; 95% CI, 0.41–5.08, *p* = 0.56.	No significant difference between women with PCOS and controls.
Choi et al., 2021 Korea [33]	Prospective, Hospital clinic	*n* = 252 (waist < 80 cm N = 212) Age 23.2 ± 5.7 BMI 22.7 ± 4.2	None	Rotterdam Clinical examination	PCOS: 2.9 years (1.5–4.5) Controls: no follow-up	Lean WC < 80	FPG HbA1c OGTT	Lean PCOS: IR T2D and prediabetes: 5.5 per 1000 PY	Increased risk of prediabetes and T2D in lean women with PCOS compared to general population. Waist circumference not associated with risk for T2D within PCOS group.
Jacewicz- Święcka et al., 2020 Poland [34]	Prospective, Outpatient clinic	*n* = 31 (*n* = 14 lean) Age 25.5 (21.5–29.2) BMI 25.6 (21.5–31.4)	None	Rotterdam Clinical examination	10 years (8.9–10.6)	Lean < 25	OGTT	No participants developed T2D at follow up. All women who developed preDM had WC ≥ 80 cm at follow up	No participants developed T2D
Velija-Asimi2016 Bosmia [35]	Prospective, Hospital clinic	*n* = 148 (*n* = 57 lean) Age 26 (21–39)	None	Rotterdam Clinical examination	3 years	Lean < 25	OGTT	0/57 lean women developed T2D at follow up.	No development of T2D in lean women with PCOS
Gambineri et al., 2012 Italy [36]	Prospective, Hospital clinic	*n* = 249 (79 lean) Age 23.4 ± 6.3 BMI 29.1 ± 7.0	None	NIH Clinical examination	16.9 years	Lean < 25	OGTT	Lean PCOS: 3/79 developed T2D Incidence rate 0.25 per 100 PY	No increased risk of T2D in lean women with PCOS compared to general population.
Andries et al., 2010 Denmark [37]	Prospective, Hospital clinic	*n* = 36 (69 PCOS and/or hirsutism, 36 lean) Age 31 (27–35) BMI 25.4 (23.0–30.0)	Non	Rotterdam Clinical examination	4 years (range 2–7)	Lean ≤ 25	OGTT	Lean PCOS T2D: 3/36	Increased risk of T2D in lean women with PCOS compared to general population.

Studies are presented in chronological order. Studies with control cohort are presented first. Values given as mean ± SD or median (min:max) or median (25–75% quartiles). Abbreviations: BMI: body mass index, FPG: fasting plasma glucose, OGTT: oral glucose tolerance test, NGT: normal glucose tolerance, IGT, impaired glucose tolerance; preDM, prediabetes; T2D, type 2 diabetes; IR, incidence rate; IRR, incidence rate ratio; PY, person-years; NS, not significant; N/A, not available.

## Data Availability

The data presented in this study are available at https://inplasy.com/inplasy-2022-6-0070/ (accessed on 16 June 2022) (code: INPLASY202260070).

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
