# Peer review of "Prospective Risk of Type 2 Diabetes in Normal Weight Women with Polycystic Ovary Syndrome"

_biomedicines, 2022, doi:10.3390/biomedicines10061455_

Round 1

Reviewer 1 Report

In this interesting paper, the authors have to review the prospective risk of T2D in normal weight of PCOS, compared with normal weight control. They have concluded that the risk of T2D could be increased in normal weight women with PCOS. They also found some individual risk markers such as hyperandrogenism, age >40 years, Asian ethnicity, and weight gain. 

1. I agree with the author that the risk of T2D in normal weight PCOS is related to ethnicity. However, in this review, the author reviewed two paper from Korea (Ryu et a. [33] and Choi et al, [36], but only one paper from Ng et al. [34]. The results from these papers did not similar, that paper from Korea showed increased T2D risk in nonobese women of PCOS, but no significant different between lean PCOS and controls. A paper from Guo et al. (Diabetes Care 2021 Jun; 44(6): e129–e130.

), has told us why the different (The Risks of Polycystic Ovary Syndrome and Diabetes Vary by Ethnic Subgroup Among Young Asian Women.)

2. I suggest the author include some paper in Chinese women with PCOS and risk of T2D.

3. PCOS women with amenorrhea or oligomenorrhea may to have severe insulin resistance than normal cycle PCOS women, and may have high risk of T2D (Panidis et al., Eur J Endocrinol 2013 Jan 17;168(2):145-52.). Lean PCOS women may have normal menstrual cycle than obese PCOS women, and therefore may have lower risk of T2D. Do you find any relationship of menstrual cycle irregularities and T2D?

Author Response

Reviewer 1:

In this interesting paper, the authors have to review the prospective risk of T2D in normal weight of PCOS, compared with normal weight control. They have concluded that the risk of T2D could be increased in normal weight women with PCOS. They also found some individual risk markers such as hyperandrogenism, age >40 years, Asian ethnicity, and weight gain. 

Reply: thank you very much for this nice comment. We have addressed suggestions below and have made adjustments in the manuscript as suggested. We would be happy to make further changes if needed.  

  1. I agree with the author that the risk of T2D in normal weight PCOS is related to ethnicity. However, in this review, the author reviewed two paper from Korea (Ryu et a. [33] and Choi et al, [36], but only one paper from Ng et al. [34]. The results from these papers did not similar, that paper from Korea showed increased T2D risk in nonobese women of PCOS, but no significant different between lean PCOS and controls. A paper from Guo et al. (Diabetes Care 2021 Jun; 44(6): e129–e130), has told us why the different (The Risks of Polycystic Ovary Syndrome and Diabetes Vary by Ethnic Subgroup Among Young Asian Women.)

Reply: thank you, we agree and have added more information regarding relevant literature in Chinese women with PCOS. The suggested paper has been added to the reference list as well.   

  1. I suggest the author include some paper in Chinese women with PCOS and risk of T2D.

Reply: thank you, we agree and have added more information regarding relevant literature in Chinese women with PCOS. The suggested paper has been added to the reference list as well.   

  1. PCOS women with amenorrhea or oligomenorrhea may to have severe insulin resistance than normal cycle PCOS women, and may have high risk of T2D (Panidis et al., Eur J Endocrinol 2013 Jan 17;168(2):145-52.). Lean PCOS women may have normal menstrual cycle than obese PCOS women, and therefore may have lower risk of T2D. Do you find any relationship of menstrual cycle irregularities and T2D?

Reply: we are grateful for the suggestion to include the aspect of menstrual cycles in the context of insulin resistance and risk of type 2 diabetes. We have added more results (lines 228-234) and a separate discussion on this issue (lines 281-287).  

Reviewer 2 Report

This review analyzed 15 studies on the potential prevalence of diabetes in lean women with PCOS. Interestingly, the attention focused on the analysis of this subgroup of patients (lean/normal weight), has made it possible to extrapolate new information on the huge debate, not yet defined, about the potential increased “risk” of T2D in these patients.

However, some points should be addressed before this manuscript may be published.

Major issues:

·       The authors should better explain the “risk” of T2D in the studies evaluated. Does it refer to the effective T2D symptoms onset as evidenced by patients’ follow-up or does it mean potential incidence of the T2D?

·       -the lack of homogeneity between the studies analyzed regarding the comparison between lean and obese patients with and without PCOS may be slightly misleading when drawing up the incidence of T2D among the various groups

·       A wider space should be reported to the discussion on the influence of the therapy in patients with PCOS and on the fact that the therapy itself could, in turn, be decisive on the risk of diabetes, regardless of whether the patient is lean or obese. In particular, the use of estrogen-progestins in an important portion of the studies in question seems to accentuate the metabolic risk of T2D, due to the negative effect of estrogens on insulin resistance.

·       Diet is another factor that should be taken into account in the discussion of the T2D symptoms onset.

·       In PCOS, autoimmunity is increased due to an increased inflammatory status. A mention should also should be added to the Hashimoto's thyroiditis which is frequent in PCOS.

Author Response

Reviewer 2:

This review analyzed 15 studies on the potential prevalence of diabetes in lean women with PCOS. Interestingly, the attention focused on the analysis of this subgroup of patients (lean/normal weight), has made it possible to extrapolate new information on the huge debate, not yet defined, about the potential increased “risk” of T2D in these patients.However, some points should be addressed before this manuscript may be published.

Reply: thank you very much for this nice comment. We have addressed suggestions below and have made adjustments in the manuscript as suggested. We would be happy to make further changes if needed.  

Major issues:

  • The authors should better explain the “risk” of T2D in the studies evaluated. Does it refer to the effective T2D symptoms onset as evidenced by patients’ follow-up or does it mean potential incidence of the T2D?

      Reply: thank you for this comment – we have specified further regarding diagnosis of T2D in included papers. The papers differed very much regarding definition of diabetes diagnosis, this had been made clearer in the Methods section lines 152-156 and Discussion line 263 (discussion section).

  • -the lack of homogeneity between the studies analyzed regarding the comparison between lean and obese patients with and without PCOS may be slightly misleading when drawing up the incidence of T2D among the various groups

      Reply: Yes, we agree and we have clearly specified the investigated study groups in the table and first part of the Discussion regarding surveillance bias.  

  • A wider space should be reported to the discussion on the influence of the therapy in patients with PCOS and on the fact that the therapy itself could, in turn, be decisive on the risk of diabetes, regardless of whether the patient is lean or obese. In particular, the use of estrogen-progestins in an important portion of the studies in question seems to accentuate the metabolic risk of T2D, due to the negative effect of estrogens on insulin resistance.

      Reply: Yes, we agree that this is a very relevant aspect. We have expanded the section regarding medical treatment including oral contraceptive treatment in the Discussion section lines 301-310

  • Diet is another factor that should be taken into account in the discussion of the T2D symptoms onset.

      Reply: We agree that diet may be important – we have emphasized this further in the discussion lines 278-389

  • In PCOS, autoimmunity is increased due to an increased inflammatory status. A mention should also should be added to the Hashimoto's thyroiditis which is frequent in PCOS.

      Reply: Yes, autoimmunity and weight gain in Hashimoto's thyroiditis could increase the risk of diabetes. We have added a section regarding this aspect in the discussion lines 306-313.  

Reviewer 3 Report

This study has the aim to review and discuss prospective risk of T2D in normal weight women with PCOS.

This is clear to me. Also that  15 studies are found  to answer the question.

The article as a whole and already in the abstract  is confusing to me : for instance in the abstract the `15 studies are mentioned but in the following it is not told “how many”  of these 15 demonstrate   a higher risk and lateron in the abstract  how many did not.

In the conclusion of the abstract I realize that your aim was not only if  there is a risk yes or no for lean women with PCOS as is said but also if more frequent follow-up must be done in these women. So you have to say that in the Aim.

You describe the 15 papers, that is fine but in the discussion you come up with facts like  depression and auto-immunity. That must be told in the introduction. In the discussion only the 15 papers must be discussed and if they answer your question, that is now in my opinion ,1. is there an increased risk and 2, must more frequent controls be done than is now normal?

In general there are said many interesting things, but that is confusing for a reader. Keep only to what you want to know and what you recommend.

Also a glossary would be helpful.

Author Response

Reviewer 3

This study has the aim to review and discuss prospective risk of T2D in normal weight women with PCOS.

This is clear to me. Also that  15 studies are found  to answer the question.

Reply: thank you very much for this nice comment. We have addressed suggestions below and have made adjustments in the manuscript as suggested. We would be happy to make further changes if needed.  

The article as a whole and already in the abstract  is confusing to me : for instance in the abstract the `15 studies are mentioned but in the following it is not told “how many”  of these 15 demonstrate   a higher risk and lateron in the abstract  how many did not.

Reply: sorry for this – we have rewritten the abstract according to suggestions and hope that you find our aim clearer. Otherwise we would be happy to make further changes.

In the conclusion of the abstract I realize that your aim was not only if  there is a risk yes or no for lean women with PCOS as is said but also if more frequent follow-up must be done in these women. So you have to say that in the Aim.

Reply: Yes of course, we have adjusted the abstract accordingly.

 You describe the 15 papers, that is fine but in the discussion you come up with facts like  depression and auto-immunity. That must be told in the introduction. In the discussion only the 15 papers must be discussed and if they answer your question, that is now in my opinion ,1. is there an increased risk and 2, must more frequent controls be done than is now normal?

Reply: We agree on this comment and have adjusted the introduction accordingly. The 15 included papers could not give a direct answer, and therefore evidence from additional literature was included. We have been more precise regarding study aims and methods. We would be happy to make further changes if requested.

In general there are said many interesting things, but that is confusing for a reader. Keep only to what you want to know and what you recommend.

Reply: Thank you very much for this comment – evidence was limited regarding risk for T2D in lean women with PCOS. Therefore, we feel that at more nuanced discussion was needed before we could give recommendations for screening and follow up. We have tried to be more specific on the context and argumentation throughout the manuscript. We have added subheadings in the discussion section to improve overview. 

Also a glossary would be helpful.

Reply: We agree and have added a glossary with relevant abbreviations. Unfortunately, the list of abbreviations for table 1 was not included in the reviewed manuscript.   

Round 2

Reviewer 1 Report

The revised manuscript is improved a lot and suitable for publication now.

Reviewer 2 Report

I confirm the interesting point emerging from this review. Authors clearly improved the manuscript that is now eligible to be published.

Reviewer 3 Report

Dear authors, I have read the revised version and agree with it. The article can be accepted in the revised version.